# Total Pain and Illness Acceptance in Pelvic Cancer Patients: Exploring Self-Efficacy and Stress in a Moderated Mediation Model

**DOI:** 10.3390/ijerph19159631

**Published:** 2022-08-05

**Authors:** Dariusz Krok, Ewa Telka, Beata Zarzycka

**Affiliations:** 1Institute of Psychology, University of Opole, 45-040 Opole, Poland; 2Department of Radiotherapy, Maria Sklodowska-Curie National Research Institute of Oncology, Gliwice Branch, 44-101 Gliwice, Poland; 3Institute of Psychology, John Paul II Catholic University of Lublin, 20-950 Lublin, Poland

**Keywords:** illness acceptance, pelvic cancer patients, perceived stress, self-efficacy, total pain

## Abstract

Cancer patients experience pain not only in its physical dimension, but also in a broader context that includes psychological, social, and spiritual aspects due to a higher level of anxiety and stress. The present prospective, longitudinal study examined the relationship between total pain and illness acceptance among pelvic cancer patients, taking into consideration the moderated mediation effects of self-efficacy and stress. The study involved a sample of pelvic cancer patients receiving radiotherapy treatment. Assessments were completed at T1 (before radiotherapy), T2 (after 3–4 weeks), and T3 (after radiotherapy) to assess the psychosocial dynamics of illness acceptance (*N* = 267). The more physical, psychological, social, and spiritual pain symptoms the patients experienced, the less they accepted negative health conditions and the effects of their illness. Stress moderated the indirect effect between total pain dimensions and illness acceptance through self-efficacy, but it did not moderate the relationship between total pain and illness acceptance. The relationships between total pain dimensions and illness acceptance thus depend on both the mediating effect of self-efficacy and the moderating effect of stress. This highlights the need to control one’s motivation and behavior and manage emotional strain or tension.

## 1. Introduction

Pain is undeniably a significant problem for cancer patients, as it affects approximately 50% at some point and nearly 80% with advanced-stage cancer [1]. Pelvic cancer patients were found to have a high level of pain and discomfort [2]. Despite growing interest in the underlying role of pain in psychological responses to cancer [3,4], relatively little attention has been paid to the role of total pain in the acceptance of the adverse consequences of the illness. We used a moderated mediation model as part of our longitudinal study to examine the connections between total pain, illness acceptance, self-efficacy, and stress in pelvic cancer patients.

### 1.1. Total Pain and Illness Acceptance

Previous research has shown that higher pain severity was related to lower illness acceptance in patients with lung cancer [5] and other types of advanced cancer [6]. Uncontrolled pain was also strongly associated with negative indicators of illness acceptance such as depression and anxiety [7]. However, there has been little empirical research on the relationship between total pain and illness acceptance.

Saunders [8,9] first proposed the concept of total pain, in which physical, psychological, social, and spiritual pain interact with one another and reflect the patient’s global sensory and emotional experience of their illness. The concept of total pain is widely accepted in the medical and psychological literature [10,11,12]; however, the lack of psychometrically validated instruments has meant that there have been few empirical studies on the relationship of total pain with illness acceptance. More recently, Raja et al. [13] proposed a new, revised definition of pain as “an unpleasant sensory and emotional experience associated with, or resembling that associated with, actual or potential tissue damage.” (p. 1980). According to this definition, pain is a multidimensional subjective experience that is influenced by biological, psychological, and social factors. It cannot be restricted to activity in sensory pathways and can have detrimental effects on social and psychological well-being and its functions. Chung et al.’s [14] qualitative study showed that the total pain intensity experienced by cancer patients was negatively related to the level of their illness acceptance. Using a case study method, Mehta and Chan [10] found that some patients with lung cancer experienced pain in its physical, social, psychological, and spiritual dimensions. In addition, the alleviation of total pain was associated with better psychosocial functioning. In another sample of cancer patients, Xu et al. [4] found that total pain acceptance was negatively associated with anxiety and depression, which are indicators of illness acceptance. In their assessment of the impact of physical activity on palliative care patients with various cancer diagnoses, Myrcik et al. [11] showed that the subjective assessment of the severity of total pain was adversely related to the quality of life. Although these studies indicated associations of total pain with illness acceptance, total pain was not measured using validated instruments that could accurately distinguish its four dimensions (physical, psychological, social, and spiritual). Furthermore, these studies primarily used qualitative data and did not examine pelvic cancer patients.

### 1.2. The Mediating and Moderating Roles of Self-Efficacy and Stress

In clinical samples, self-efficacy was found to be related to both pain [15] and illness acceptance [16], suggesting that it may play a mediating role in the above-mentioned relationship. Using structured equation modeling (SEM) analysis, Hirai et al. [17] discovered that self-efficacy was a mediator between physical condition (which also included the dimensions of pain experience) and psychological adjustment to illness among advanced cancer patients. In particular, patients in good physical condition had a high level of self-efficacy that was linked to improved psychological adjustment. After studying women with breast cancer, Haas [18] demonstrated that self-efficacy mediated the association between fatigue (which also comprised pain as an additional symptom) and physical well-being but not the influence of fatigue on quality of life. Self-efficacy also mediated the impact of symptom severity (which included items measuring pain) on the quality of life in patients with different forms of cancer; self-efficacy played a positive role here, as it was linked to a better quality of life [19]. Self-esteem, but not self-efficacy, mediated the associations between pain intensity and health-related quality of life in a group of teenagers with chronic pain [20]. This raises the question of whether self-efficacy might play a mediating role in pelvic cancer patients who tend to experience specific types of pain and discomfort resulting from the cancer’s physiological and psychosocial components [2].

Because cancer patients are exposed to severe stress, feelings of emotional strain or tension resulting from adverse and traumatic illness-related conditions are likely to have an impact on the aforementioned relationships. Previous studies demonstrated the interaction between stress and pain [21], self-efficacy, and illness acceptance [22] in cancer patients as a result of its strong psychosocial impact. Examining children with cancer, Aldridge and Roesch [23] found that the level and form of stress moderated the relationship between the child’s cognitive appraisals and psychological and physical adjustment to illness. In a sample of breast cancer survivors, perceived stress was shown to moderate the relationship between person-centered inflammation, which included pain as one of the components, and depressive symptoms; the effects were stronger among patients who reported high stress than patients who reported low stress [24].

Although these studies have substantially increased our understanding of the psychological consequences of pain, no single study examining total pain, illness acceptance, self-efficacy, and stress has been conducted, including among pelvic cancer patients. The complex nature of the relationships between the above variables may be better understood using Secinti et al.’s [25] integrated model of cancer acceptance. According to its assumptions, acceptance of cancer involves behavioral activity in response to cancer-related stress resulting from internal experiences of the illness, such as physiological sensations, cognitive appraisals, and emotional reactions. In attempting to regulate these internal experiences, individuals can develop an attitude of either acceptance or denial of the illness. In the first instance, this will make it easier for them to cope with the illness, but in the second, it will cause them to suffer more. We may assume, drawing on the model, that the relationship between the experience of total pain and illness acceptance will depend on the individual’s belief in his or her ability to deal successfully with the illness. As self-efficacy reflects behavioral activities, it was chosen as a proxy for behavioral strategies for coping with pain, cancer, and treatments, which would lead to greater acceptance. Furthermore, cancer-related stress may modify this process by limiting the individual’s ability to act and adjust to the disease.

### 1.3. The Present Study

No research has directly examined the mediating role of self-efficacy and the moderating role of stress in the context of total pain and illness acceptance among pelvic cancer patients, leaving a gap in the existing empirical knowledge of these interactions. The present study aimed to apply a moderated mediation model to investigate the associations between total pain (T1—before radiotherapy), self-efficacy, stress (T2—approximately halfway through the overall radiotherapy, i.e., after 3–4 weeks), and illness acceptance (T3—after radiotherapy, i.e., after 6–8 weeks) in pelvic cancer patients undergoing radiotherapy. The variables were measured at three separate time points to examine their temporal relations for a prospective moderated mediation model [26] (Figure 1).

We developed four hypotheses based on the integrated model of acceptance of cancer and previous research:

**H1.** *Total pain dimensions will be negatively related to illness acceptance*;

**H2.** 
*Self-efficacy will mediate the relationship between total pain dimensions and illness acceptance; higher pain will lead to lower illness acceptance through the decrease of self-efficacy;*


**H3.** 
*Stress moderates the indirect effect between total pain dimensions and illness acceptance through self-efficacy; the positive effect is stronger when stress is low vs. high;*


**H4.** 
*Stress moderates the direct effect between total pain dimensions and illness acceptance through self-efficacy; the positive effect is stronger when stress is low vs. high.*


## 2. Materials and Methods

### 2.1. Power Analysis

The power analysis recommended by Preacher et al.’s [27] for moderated mediation models was used to secure a sufficient sample size. In all the variables analyzed, a sample size of *n* = 200 or more was sufficient to obtain an effect size of *p* = 0.05 and a statistical power > 0.80. To avoid Type II errors, the bootstrapped samples in the PROCESS macro were set to 5000 with 95% bias-corrected confidence intervals, which was statistically adequate for the number of respondents.

### 2.2. Participants and Recruitment Procedure

Patients with pelvic cancer were recruited at the National Research Institute of Oncology in Gliwice and the Oncological Hospital in Opole between October 2019 and November 2021. They were approached by research assistants and asked if they would like to participate in the study. The following criteria were required for inclusion: a confirmed diagnosis of pelvic cancer, completion of radiotherapy treatment, and a positive prognosis for recovery. The exclusion criteria: inability to participate due to other medical reasons (e.g., visual impairment and extreme fatigue), serious major psychiatric disorders, or cognitive deficit. Participants who volunteered to take part in the study were given a questionnaire and an informed consent form and asked to return them to the researcher within 1 week. Three-hundred-and-eighty-four patients were enrolled and attended the T1 (pre-radiotherapy) interview; 311 patients attended the T2 (approximately halfway through radiotherapy) interview, and 267 patients (141 women and 126 men) completed the T3 (at the end of radiotherapy) interview. The final participation rate was 69.53%. Patients were debriefed upon completion of the study and provided contact information in case the researchers had any further questions. The University of Opole Institutional Ethics Committee approved the study (approval number 4/2019).

### 2.3. Measures

The Total Pain Questionnaire (TPQ) was used to determine total pain [28]. It is based on the theoretical concept of total pain [8,9], and refers to the patient’s pain as distressing, painful feelings experienced in four main dimensions: physical, psychological, social, and spiritual. The TPQ comprises 20 items that are grouped into four subscales: (a) physical pain, that is, unpleasant and discomforting physical sensations from known tissue injury (“I experience unpleasant and painful physical sensations related to the illness” or “I feel pain in different parts of my body”); (b) psychological pain, that is, subjective feelings of anxiety, depression, and hopelessness in the context of medical uncertainty (“Illness is a source of negative thoughts and feelings” or “I experience a sense of helplessness and anxiety because of my illness”; (c) social pain, that is, the experience of not being able to play a role, losing social status or a job, fears for the future of one’s family, a fear of being dependent on others, or financial anxiety (“Being dependent on other people causes me pain” or “I feel lonely and abandoned by other people”); and (d) spiritual pain, that is, negative feelings of disconnection and abandonment associated with spiritual/religious beliefs (“My faith (e.g., in God or a higher power) has weakened due to the illness” or “I feel an inner, spiritual suffering”). The confirmatory factor analysis showed satisfactory goodness of fit indices of the four-factor model: χ^2^ (*N* = 523) = 215.98, *p* < 0.001; NFI = 0.91; CFI = 0.92; RMSEA = 0.05; SRMR = 0.04. The sum of the scores obtained for all items produces a global score that measures total pain. The Cronbach’s alpha coefficients were 0.85 for physical pain, 0.88 for psychological pain, 0.82 for social pain, 0.80 for spiritual pain, and 0.89 for the total score.

The Perceived Stress Scale (PSS-10) [29] was used to assess stress. This widely used scale assesses cancer patients’ perceived stress levels and comprises ten items rated on a 5-point Likert scale, from 0 (never) to 4 (very often). A higher score on the scale indicates a higher level of perceived stress. Juczyński and Ogińska-Bulik’s [30] translation was used. The Cronbach’s alpha coefficient was 0.79.

The General Self-Efficacy Scale (GSES) was used to measure self-efficacy. It assesses a general sense of self-efficacy that reflects one’s ability to deal with challenging or novel situations successfully [31]. The scale consists of ten items rated on a 4-point Likert scale ranging from 1 (strongly disagree) to 4 (strongly agree). The higher score on the scale, the higher level of general self-efficacy. Schwarzer, Jerusalem, and Juczyński’s [32] Polish version was used, and Cronbach’s alpha coefficient was 0.89.

The Acceptance of Life with the Disease Scale, which estimates one’s adaptation to the disease in terms of the ability to accept health conditions and maintain overall life satisfaction [33], was used to assess illness acceptance. It comprises 20 items rated on a 4-point scale ranging from 1 (no) to 4 (yes). The measure has three subscales: (a) satisfaction with life; (b) reconciliation with the disease; and (c) self-distancing from the disease. The sum of the subscale scores yielded the total score, and Cronbach’s alpha coefficient was 0.91.

Family support. In addition, we assessed the level of support received from family members. It was measured by one item “How much support do you receive from your family members” scored on a 10-point Likert scale, ranging from 1 (not at all) to 10 (a lot).

### 2.4. Data Analysis

First, G*Power software was used to determine the minimum sample size *N* required within the model. The following conditions were applied a priori: a significance threshold of α = 0.05 and recommended test power (1 − β) = 0.90 [34]. The minimum sample size was estimated to be *N* = 260 participants. In addition, as all the variables were measured by self-report questionnaires, we decided to apply Harman’s single factor test to rule out the possibility of common method bias (CMB) [35]. The test revealed that a single factor explained 20.49% of the variance and, in addition, all the items were independent dimensions. These results prove that our data were free of common method bias and accurately reflected the population of pelvic cancer patients in our country. The variables were measured at the T1 (total pain), T2 (self-efficacy, stress), and T3 (illness acceptance) interviews. Mediation analysis was used to examine whether self-efficacy may have mediated the association between the four total pain variables and their total score with illness acceptance (Model 4; 95% confidence intervals with 5000 bootstrap samples). Moderated mediation analysis was used to assess whether stress may have moderated the indirect effects between total pain and its dimensions and illness acceptance through self-efficacy (Model 8; 95% confidence intervals with 5000 bootstrap samples) [26].

## 3. Results

### 3.1. Correlational Analysis

Age was found to have negative associations with psychological pain, social pain, spiritual pain, total pain (total score), and stress in the initial analysis. However, it was positively associated with illness acceptance. All the dimensions of total pain and its total score were negatively associated with self-efficacy and illness acceptance. By contrast, the dimensions of total pain and its total score were positively associated with stress. No associations were found between self-efficacy and stress. In addition, family support negatively correlated with psychological, social, and total pain, and stress, but positively correlated with self-efficacy (Table 1).

### 3.2. Mediation Analysis

Direct effects demonstrated that physical pain, psychological pain, social pain, spiritual pain, and the total score were negatively associated with self-efficacy and illness acceptance (Table 2). By contrast, self-efficacy was positively associated with illness acceptance for all kinds of pain. Indirect effects revealed that self-efficacy was a mediator between total pain and its four dimensions (i.e., physical, psychological, social, and spiritual) and illness acceptance. Total pain and its dimensions were associated with lower self-efficacy, which in turn was associated with higher illness acceptance.

Another intriguing result emerged from the comparison of mediation effect-size measures (*R*^2^), which assesses variance in mediation models and shows the proportion of the total effect mediated by the intervening variable [36]. The indirect path of social pain—self-efficacy—illness acceptance (0.10) had the largest effect, while the indirect path of physical pain—self-efficacy—illness acceptance (0.03) had the weakest. The mediating power of self-efficacy for social pain was, therefore, three times greater than it was for physical pain.

### 3.3. Moderated Mediation Analysis

For physical, social, spiritual, and total pain, the interaction between pain and stress for self-efficacy (Interaction 1) as an outcome variable was significant, but not for psychological pain (Table 3). By contrast, for all the dimensions of pain and its total score, the interaction between pain and stress for illness acceptance (Interaction 2) as an outcome variable was nonsignificant. In cases of psychological, social, and total pain, the conditional direct effects were stronger for patients with low stress than for those with high stress, though their signs were reversed. The effects were nonsignificant for physical pain and spiritual pain.

The index of moderated mediation was significant for physical, social, spiritual, and total pain, but not for psychological pain, which suggests that stress moderated the indirect effect of total pain and its physical, social, spiritual dimensions with illness acceptance through self-efficacy. Not all conditional indirect effects were significant. For social, spiritual, and total pain, the conditional indirect effects for patients with low stress were stronger than for those with high stress; however, their signs were reversed. These effects did not occur in the case of physical pain (due to a non-significant interaction between high stress and physical pain) and psychological pain (due to the above-mentioned nonsignificant interaction between pain and stress). For total pain, a comparison of the mediation and moderated mediation models revealed that 5% of the variance was explained by illness acceptance (Δ*R*^2^ = 0.05).

## 4. Discussion

The present study tested a complex model in which resilience was a mediator and stress was a moderator in the relationship between total pain with illness acceptance in a sample of pelvic cancer patients undergoing radiotherapy. Because pelvic cancer patients experience a relatively high level of discomfort and soreness [2], it is critical to identify psychological factors that can increase illness acceptance and the underlying mechanisms of such effects. To the best of our knowledge, the moderated mediation effects among pelvic cancer patients have not hitherto been investigated in a longitudinal, prospective study.

### 4.1. The Relationship between Total Pain and Illness Acceptance

Our findings demonstrate that the dimensions of total pain were negatively associated with a person’s illness acceptance, thus confirming Hypothesis 1. In other words, the more the patients experienced physical, psychological, social, and spiritual symptoms of pain, the less they accepted their negative health conditions and expressed satisfaction with their lives. This result confirms previous theoretical speculations that people often experience painful symptoms in a holistic fashion that extends beyond the physiological domain [10,12] to reflect their psychological, social, and cultural characteristics. Furthermore, people’s ability to accept their health condition will depend on these holistic aspects. The experience of pain is frequently global in nature and entails the interplay of physiological, psychological, social, and spiritual aspects to shape a unique human experience. Our findings also reflect the interactive nature of total pain components and suggest that physical pain may exacerbate painful psychological, social, and spiritual symptoms from a perceived sense of anxiety, interpersonal abandonment, and meaninglessness.

### 4.2. The Mediating Roles of Self-Efficacy

Hypothesis 2 stated that self-efficacy played a role in the relationship between total pain and illness acceptance. Our findings show that self-efficacy was a mediator between total pain and its four dimensions and disease acceptance, so the hypothesis was confirmed. In particular, total pain and its dimensions were related to lower self-efficacy, which, in turn, was linked to increased disease acceptance. Previous research has shown that self-efficacy mediated associations between symptom severity and well-being among cancer patients [19] and between self-reported pain and functional limitation in women suffering from fibromyalgia [37]. Our results complement the existing literature by implying that the interplay of different dimensions of pain with illness acceptance is the result of a person’s capacity to manage difficult situations, allowing pelvic cancer patients to accept the pressures of their illness and maintain an optimal level of life satisfaction. One explanation for this mechanism is that resilient individuals are better equipped psychologically to manage their fears, avoid denial, and demonstrate individual and social competence, all of which allows them to cope with stress and emergent life challenges [22].

The strongest mediation effect was shown for social pain, whereas the weakest effect was shown for physical pain. This implies that the patients’ capacity to meet potential challenges and reach health-related goals in order to accept the illness is particularly important in the context of fears and anxieties regarding family and social relationships, reliance on others, and financial worries. Previous studies have suggested that the social aspects of pain experienced by some cancer patients may affect their quality of life more than physical aspects due to an illness-related reduced ability to fulfill social roles associated with family, financial status, or professional obligations [10,38]. The ability to meet social expectations, on the other hand, appears to be an important mediating factor in minimizing the negative impact of social pain on illness acceptance.

### 4.3. The Moderated Mediation Effects of Stress and Self-Efficacy

Our results largely provide support to the notion that stress moderates the indirect effect between total pain dimensions and illness acceptance through self-efficacy; Hypothesis 3 was therefore confirmed. Stress was a moderator for physical, social, and spiritual pain, but not for psychological pain; stress also had a moderated effect on total pain. The mediational effects of self-efficacy were stronger for patients with low stress than for those with high stress. These results are consistent with studies that have shown the moderating effect of stress on the relationship between painful feelings and mental adjustment to illness [23,24]. However, the present study examined the moderating role of stress over time between total pain dimensions and illness acceptance through self-efficacy, making for more dynamic analysis. Stress should not be viewed as an autonomous biological reaction affecting illness acceptance, but rather as a factor that interacts with different facets of total pain.

The moderated mediation effects of stress and self-efficacy are better understood within the integrated model of acceptance of cancer [25] when the interactive character of emotional and motivational factors is highlighted. Consistent with the model, pelvic cancer patients who experienced severe stress due to painful physical symptoms (physical pain) and adverse social-existential appraisals (social and spiritual pain) were less able to perform daily activities and complete life tasks, which, lead to a lower level of illness acceptance. The ability to mitigate, if only partially, such negative total pain experiences allows patients to lower their stress levels and improve their ability to manage illness-related situations, which in turn will improve their capacity to adjust to their illness. This interpretation is supported by Mosher et al.’s [39] study, which found that more activities directed at overcoming the negative consequences of cancer and the acceptance of cancer were associated with fewer symptoms of pain.

The present study could not confirm Hypothesis 4 (that stress moderated the direct effect between total pain dimensions and illness acceptance) because the respective variables showed no interaction. Previous research suggested that stress was a moderator of the association between painful appraisals and adjustment to illness in children with cancer [23] or painful inflammation and depressive symptoms in breast cancer survivors [24], but our findings could not confirm this. The lack of moderation effects does not necessarily invalidate earlier cross-sectional findings, as it seems feasible that the previous studies described more correlational mechanisms in their samples, or that the relationships may change over time due to potential fluctuations in total pain experiences and ensuing stress levels [40]. The inconclusive findings point to the need for further research into the complex relationships between pain, stress, and illness acceptance.

### 4.4. Strengths and Limitations

The present study has several strengths. First, it is the first empirical investigation we are aware of that examines the concept of total pain and its four dimensions in relation to illness acceptance among pelvic cancer patients. It provided a very rare opportunity to observe complex longitudinal relationships between the different elements of total pain and other psychologically important factors. Second, we used a longitudinal design to look at the directionality between variables within our moderated mediation model [25]. Third, the assessments of stress and self-efficacy were conducted at the same point in time (T2), allowing for an analysis of their moderated mediation effects. The present study also has several limitations. First, the sample comprised pelvic cancer patients, providing insight into the total pain and illness acceptance mechanisms of a medically specific population. As a result, generalizing the results to other cancer patient populations should be carried out cautiously [3]. Second, the dropout from T1 to T3 may have indicated sampling bias and confined the generalizability of the findings to patients who completed the longitudinal assessment. Third, we did not have sufficient capacity to monitor differences in social and family support, which might have impacted the psychological and social dimensions of total pain. Fourth, as most of the scales used were not specific to pelvic cancer situations, our results can be applied to other groups of cancer patients with a high degree of caution. Indeed, biological and psychological conditions that are specific to different types of cancer may modify the results. Such limitations should be considered in future research.

## 5. Conclusions

The present study identified the underlying mechanisms in the association between total pain dimensions with illness acceptance in pelvic cancer patients, which are part of the moderated mediation effects of self-efficacy and stress. Overall, the findings demonstrate the need to examine pain in its entirety, including physical, psychological, social, and spiritual dimensions, as well as psychosocial factors such as self-efficacy and stress. Finally, our research provides important information and guidelines for intervention programs to alleviate pain in cancer patients. They may include developing education programs aimed at (1) enhancing cognitive skills and self-efficacy for communicating with medical staff about pain and (2) teaching patients how to manage stress effectively by monitoring pain intensity, which uses multidimensional scales that assess pain in its physical, psychological, social, and spiritual dimensions, and (3) promoting materials that emphasize psychological and spiritual aspects of pain experiences.

## Figures and Tables

**Figure 1 ijerph-19-09631-f001:**
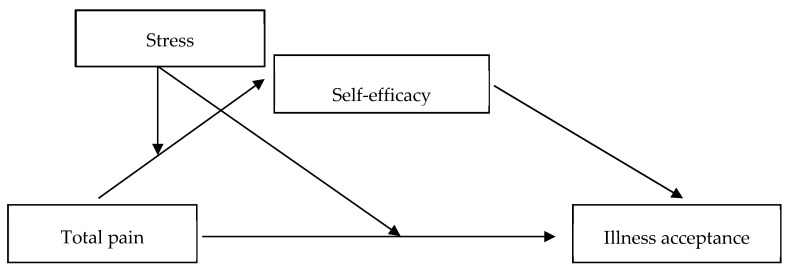
The general moderated mediation model.

**Table 1 ijerph-19-09631-t001:** Means, standard deviations, and correlations among total pain dimensions, self-efficacy, stress, and illness acceptance.

Variables	*M*	*SD*	1.	2.	3.	4.	5.	6.	7.	8.	9.
Age	61.26	12.77	—								
2. Physical pain	3.21	2.42	−0.10	—							
3. Psychological pain	3.74	2.42	−0.13 *	0.57 ***	—						
4. Social pain	3.05	2.29	−0.18 **	0.61 ***	0.74 ***	—					
5. Spiritual pain	3.15	2.21	−0.12 *	0.59 ***	0.69 ***	0.81 ***	—				
6. Total pain	3.29	2.02	−0.15 *	0.80 ***	0.87 ***	0.81 ***	0.84 ***	–			
7. Self-efficacy	3.01	0.53	0.11	−0.15 *	−0.26 ***	−0.33 ***	−0.29 ***	−0.29 ***	–		
8. Stress	2.71	0.98	−0.18 **	0.42 ***	0.38 ***	0.51 ***	0.44 ***	0.50 ***	−0.05	–	
9. Illness acceptance	2.88	0.50	0.18 **	−0.24 ***	−0.40 ***	−0.41 ***	−0.33 ***	0.40 ***	0.48 ***	−0.28 ***	
10. Family support	4.58	3.41	0.07	−0.08	−0.29 ***	−0.32 ***	−0.11	−0.20 **	0.26 ***	−0.21 **	0.08

* *p* < 0.05; ** *p* < 0.01; *** *p* < 0.001.

**Table 2 ijerph-19-09631-t002:** Mediation estimates for self-efficacy in mediating the relationship between total pain and illness acceptance (standardised coefficients and effects).

Variables	*B*	*SE*	*t [* *LLCI, ULCI]*	Model *R*^2^
Direct effects				
Physical pain–Self-efficacy	−0.14	0.01	−2.40 [−0.06, −0.01]	0.02 *
Self-efficacy–Illness acceptance	0.46	0.05	8.61 [0.34, 0.53]	
Physical pain–Illness acceptance	−0.17	0.01	−3.18 [−0.06, −0.01]	0.26 ***
Psychological pain–Self-efficacy	−0.26	0.01	−4.42 [−0.08, −0.03]	0.07 ***
Self-efficacy–Illness acceptance	0.41	0.05	7.71 [0.29, 0.48]	
Psychological pain–Illness acceptance	−0.29	0.01	−5.54 [−0.08, −0.04]	0.31 ***
Social pain–Self-efficacy	−0.33	0.01	−5.64 [−0.10, −0.05]	0.11 ***
Self-efficacy–Illness acceptance	0.39	0.05	7.22 [0.27, 0.47]	
Social pain–Illness acceptance	−0.28	0.01	−5.23 [−0.09, −0.04]	0.31 ***
Spiritual pain–Self-efficacy	−0.28	0.02	−4.92 [−0.10, −0.04]	0.08 ***
Self-efficacy–Illness acceptance	0.43	0.05	7.76 [0.30, 0.51]	
Spiritual pain–Illness acceptance	−0.20	0.01	−3.70 [−0.07, −0.02]	0.27 ***
Total pain-Self–efficacy	−0.29	0.02	−5.01 [−0.11, −0.05]	0.09 ***
Self-efficacy–Illness acceptance	0.38	0.05	7.50 [0.28, 0.48]	
Total pain–Illness acceptance	−0.28	0.01	−5.18 [−0.09, −0.04]	0.30 ***
Indirect effect	*Effect*	*SE*	*LLCI*	*ULCI*
Physical pain–Self-efficacy–Illness acceptance	−0.07	0.03	−0.13	−0.01
Psychological pain–Self-efficacy–Illness acceptance	−0.11	0.03	−0.17	−0.05
Social pain–Self-efficacy–Illness acceptance	−0.13	0.03	−0.20	−0.08
Spiritual pain–Self-efficacy–Illness acceptance	−0.12	0.04	−0.20	−0.06
Total pain–Self-efficacy–Illness acceptance	−0.12	0.03	−0.18	−0.06
*R*^2^ mediation effect size				
Physical pain–Self-efficacy–Illness acceptance	0.03	0.02	0.01	0.07
Psychological pain–Self-efficacy–Illness acceptance	0.08	0.03	0.03	0.14
Social pain–Self-efficacy–Illness acceptance	0.10	0.03	0.05	0.16
Spiritual pain–Self-efficacy–Illness acceptance	0.07	0.02	0.03	0.13
Total pain–Self-efficacy–Illness acceptance	0.08	0.03	0.04	0.15

*** *p* < 0.001; * *p* < 0.05.

**Table 3 ijerph-19-09631-t003:** Moderated mediation estimates for illness acceptance outcomes.

Variables	*B*	*SE*	*t [LLCI, ULCI]*	
INTERACTIVE EFFECTS				
*Physical pain as independent variable*				
Interaction 1: Physical pain × Stress	0.04	0.01	2.96 [0.01, 0.07]	
Interaction 2: Physical pain × Stress	−0.01	0.01	−0.21 [−0.02, 0.02]	
*Psychological pain as independent variable*				
Interaction 1: Psychological pain × Stress	0.02	0.01	1.10 [−0.01, 0.04]	
Interaction 2: Psychological pain × Stress	0.02	0.01	1.63 [−0.01, 0.03]	
*Social pain as independent variable*				
Interaction 1: Social pain × Stress	0.04	0.01	2.61 [0.01, 0.06]	
Interaction 2: Social pain × Stress	0.01	0.01	1.08 [−0.01, 0.04]	
*Spiritual pain as independent variable*				
Interaction 1: Spiritual pain × Stress	0.04	0.01	2.75 [0.01, 0.07]	
Interaction 2: Spiritual pain × Stress	0.02	0.01	1.24 [−0.01, 0.03]	
*Total pain as independent variable*				
Interaction 1: Total pain × Stress	0.04	0.01	2.62 [0.01, 0.07]	
Interaction 2: Total pain × Stress	0.02	0.01	1.12 [−0.01, 0.04]	
CONDITIONAL DIRECT EFFECTS	*Effect*	*SE*	*LLCI*	*ULCI*
Low stress × Physical pain	−0.01	0.01	−0.05	0.02
High stress × Physical pain	−0.01	0.01	−0.04	0.01
Low stress × Psychological pain	−0.07	0.02	−0.10	−0.03
High stress × Psychological pain	−0.03	0.02	−0.06	−0.01
Low stress × Social pain	−0.06	0.02	−0.10	−0.02
High stress × Social pain	−0.03	0.02	−0.07	−0.01
Low stress × Spiritual pain	−0.04	0.02	−0.08	−0.01
High stress × Spiritual pain	−0.01	0.02	−0.04	0.02
Low stress × Total pain	−0.06	0.02	−0.10	−0.02
High stress × Total pain	−0.04	0.02	−0.07	−0.01
CONDITIONAL INDIRECT EFFECTS	*Effect*	*SE*	*LLCI*	*ULCI*
Low stress × Physical pain	−0.03	0.01	−0.06	−0.01
High stress × Physical pain	0.01	0.01	−0.02	0.02
Low stress × Psychological pain	−0.03	0.01	−0.05	−0.01
High stress × Psychological pain	−0.02	0.01	−0.04	−0.01
Low stress × Social pain	−0.05	0.01	−0.08	−0.03
High stress × Social pain	−0.02	0.01	−0.05	−0.01
Low stress × Spiritual pain	−0.05	0.01	−0.08	−0.02
High stress × Spiritual pain	−0.02	0.01	−0.04	−0.01
Low stress × Total pain	−0.05	0.01	−0.08	−0.02
High stress × Total pain	−0.03	0.01	−0.05	−0.01
INDEX OF MODERATED MEDIATION				
Physical pain as independent variable	0.02	0.01	0.01	0.04
Psychological pain as independent variable	0.01	0.01	−0.01	0.02
Social pain as independent variable	0.02	0.01	0.001	0.02
Spiritual pain as independent variable	0.02	0.01	0.01	0.03
Total pain as independent variable	0.02	0.01	0.005	0.03

## Data Availability

The empirical data for this article can be found at the OSF HOME repository, https://osf.io/9k2dj/ (accessed on 3 August 2022).

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
