# Peer review of "Total Pain and Illness Acceptance in Pelvic Cancer Patients: Exploring Self-Efficacy and Stress in a Moderated Mediation Model"

_ijerph, 2022, doi:10.3390/ijerph19159631_

Round 1
Reviewer 1 Report
Main comment:
The article focus on the relationship between (total) pain and illness acceptance, with a cancer pain populations, hypothesizing that general self-efficacy is a mechanism through which the first relationship unfolds. Furthermore, they look to confirm if the indirect effect of self-efficacy on the negative relationship between total pain and illness acceptance was conditioned to perceived stress level. The methods and data analysis are aligned with the research goals.
Specific comments:
1) Pain is a multidimensional experience and – without disregard for Saunders (1964; 1967) – more recent references exist that could be included (Srinivase et al 2020) doi: 10.1097/j.pain.0000000000001939
2) Page 1, line 15 “the moderation moderated mediation effects” confuses the reader.
3) Still in the abstract, lines 23 to 25 authors should rephrase into a more comprehensive style so a broader audience can understand what the results were.
4) I understand that the authors picked up on Secinti et al model to choose self-efficacy for a mediator, maybe as a proxy for behavioral strategies for cope with pain, cancer and treatments which would lead to greater acceptance. My interrogation lays in why a behavioral variable (even self-reported) was not included.
5) Moreover, the hypotheses for the indirect effect should be more specific, hypothesizing the direction of the relationships. For example, higher pain would lead to lower illness acceptance through the decrease of self-efficacy?
6) Most of the scales used were not specific for pelvic cancer situations, authors should reflect on how this might have influenced the results and include it as a limitation.
7) Discussion, page 8, line 292: “resilience”?
8) Page 11, line 412-414: authors could provide an example of what insight they take out for future intervention programs to alleviate pain in cancer patients.
Author Response
Responses to Reviewer 1
Dear Reviewer,
We would like to thank you for the words of appreciation and your comments regarding the manuscript. We appreciate your thoughtfulness and time spent on presenting your concerns regarding our research. Based on your comments we revised the paper. Additionally, we responded to all of the issues raised during the revision process. Below we present detailed responses to all your comments.
Thank you once again for helping us to improve the quality of our research.
Specific comments:
1) Pain is a multidimensional experience and – without disregard for Saunders (1964; 1967) – more recent references exist that could be included (Srinivase et al 2020) doi: 10.1097/j.pain.0000000000001939
– In line with this suggestion, we added more information about pain being multidimensional experience and more recent references. We stated that more recently, Raja et al. [13] proposed a new, revised definition of pain as “an unpleasant sensory and emotional experience associated with, or resembling that associated with, actual or potential tissue damage.” (p. 1980). According to this definition, pain is a multidimensional subjective experience that is influenced by biological, psychological, and social factors. It cannot be restricted to activity in sensory pathways and can have detrimental effects on social and psychological well-being and its functions.
2) Page 1, line 15 “the moderation moderated mediation effects” confuses the reader.
– It is a misleading phrase, indeed. Inadvertently, the word “moderation” is a typo and the proper phrase should be” The moderated mediation effects”. We corrected it in our revision. Therefore, the current sentence is: “The present prospective, longitudinal study examined the relationship between total pain and illness acceptance among pelvic cancer patients, taking into consideration the moderated mediation effects of self-efficacy and stress.”
3) Still in the abstract, lines 23 to 25 authors should rephrase into a more comprehensive style so a broader audience can understand what the results were.
– The lines 23 to 25 were rephrased to more clearly reflect the results obtained in our study. The new sentence is: “The relationships between total pain dimensions and illness acceptance thus depend on both the mediating effect of self-efficacy and the moderating effect of stress.”
4) I understand that the authors picked up on Secinti et al model to choose self-efficacy for a mediator, maybe as a proxy for behavioral strategies for cope with pain, cancer and treatments which would lead to greater acceptance. My interrogation lays in why a behavioral variable (even self-reported) was not included.
– Thank you for this comment as we initially thought about including some self-reported behavioural variable, but finally decided to use self-efficacy as a main measure. The key reason was that self-efficacy reflects behavioural activities, since it refers to an individual's belief in his or her capacity to execute behaviours necessary to produce specific performance attainments. In other words, self-efficacy beliefs reflect the domain of functioning and circumstances related to one’s behaviour. Therefore, we thought that using self-efficacy would represent the behavioural component included in Secinti et al.’s model. As the reviewer stated, we then chose self-efficacy as a proxy for behavioural strategies for coping with pain, cancer and treatments which would lead to greater acceptance. We used Secinti et al.’s model to provide a theoretical framework for conceptualising relationships among total pain, illness acceptance, self-efficacy, and stress. Within the model, acceptance of cancer involves behavioural actions aligned with deeply held values, so it is indeed a coping strategy (i.e. reflecting one’s behaviour).
We added the explanation regarding the behavioural character of self-efficacy in our revision.
5) Moreover, the hypotheses for the indirect effect should be more specific, hypothesizing the direction of the relationships. For example, higher pain would lead to lower illness acceptance through the decrease of self-efficacy?
– Hypothesizing the direction of the relationships is needed, indeed. Therefore, we specified our hypothesis so that they would reflect the directionality. The altered hypotheses are:
H2: self-efficacy will mediate the relationship between total pain dimensions and illness acceptance; higher pain will lead to lower illness acceptance through the decrease of self-efficacy;
H3: stress moderates the indirect effect between total pain dimensions and illness acceptance through self-efficacy; the positive effect is stronger when stress is low vs high;
H4: stress moderates the direct effect between total pain dimensions and illness acceptance through self-efficacy; the positive effect is stronger when stress is low vs. high.
6) Most of the scales used were not specific for pelvic cancer situations, authors should reflect on how this might have influenced the results and include it as a limitation.
– In the limitation section, we added the point that most of our scales were not specific for pelvic cancer situations, and as a consequence , our results can be applied to other groups of cancer patients with a high degree of caution. Indeed, biological and psychological conditions that are specific to different types of cancer may modify the results.
7) Discussion, page 8, line 292: “resilience”?
– The sentence on p. 8, lines 292-294 is definitely imprecise with regard to the phrase: “a model of resilience” as it may imply that we primarily tested a model of resilience. Therefore, we altered the sentence so that it would clearly reflect the aim of our study. The new sentence is: “The present study tested a complex model in which resilience was a mediator and stress was a moderator in the relationship of total pain with illness acceptance in a sample of pelvic cancer patients undergoing radiotherapy.”
8) Page 11, line 412-414: authors could provide an example of what insight they take out for future intervention programs to alleviate pain in cancer patients.
– In Conclusions, we added information regarding potential interventions to alleviate pain in cancer patients. We specified that the interventions can include developing education programs aimed at: (1) enhancing cognitive skills and self-efficacy for communicating with medical staff about pain and (2) teaching patients how to manage stress effectively by monitoring pain intensity, which uses multidimensional scales that assess pain in its physical, psychological, social, and spiritual dimensions, and (3) promoting materials that emphasize psychological and spiritual aspects of pain experiences.

Reviewer 2 Report
The paper is coherent to the aim of the journal, however as admitted y the authors themselves there several limitations that according
"The present study also has several limitations. First, the sample comprised pelvic cancer patients, providing insight into the total pain and illness acceptance mecha-nisms of a medically specific population. As a result, generalizing the results toother cancer patient populations should be carried out cautiously [3]. Second, dropout from T1 to T3 may have indicated sampling bias and confined the generalizability of the findings to patients who completed the longitudinal assessment. Third, we did not have sufficient capacity to monitor differences in social and family support, which might have impacted the psychological and social dimensions of total pain."
I think that the third weakness could have a real impact to the paper so I suggest the author to try to correct this bias if possible.
Reference and structure of the paper are correct.
Author Response
Responses to Reviewer 2
We would like to thank the Reviewer for the words of appreciation and the feedback on our manuscript. We have made changes in response to your comments/recommendations. This process has strengthened the manuscript. Please see below our responses to the reviewer’s feedback.
Specific comments:
The paper is coherent to the aim of the journal, however as admitted y the authors themselves there several limitations: "The present study also has several limitations. First, the sample comprised pelvic cancer patients, providing insight into the total pain and illness acceptance mecha-nisms of a medically specific population. As a result, generalizing the results toother cancer patient populations should be carried out cautiously [3]. Second, dropout from T1 to T3 may have indicated sampling bias and confined the generalizability of the findings to patients who completed the longitudinal assessment. Third, we did not have sufficient capacity to monitor differences in social and family support, which might have impacted the psychological and social dimensions of total pain."
I think that the third weakness could have a real impact to the paper so I suggest the author to try to correct this bias if possible. Reference and structure of the paper are correct.
– While devising our research we did not unfortunately take into account direct measures of social and family support, which indeed would be highly valuable. We realise that social and family support plays a significant role in cancer patient’s illness acceptance and abilities to cope with pain. Both forms of support can significantly help patients in dealing with pain associated with cancer and its acceptance. Nonetheless, we did not intend to analyse it in our project, instead, we concentrated on total pain, illness acceptance, self-efficacy, and stress.
The only factor that we included in our project was a level of family support that was measured by one item “How much support do you receive from your family” scored on a 10-point Likert scale, ranging from 1 (not at all) to 10 (a lot). Initially, we did not include the answers in our original manuscript as the aim of our study was not to examine social or family support. However, as the Reviewer suggested that some information regarding social and family support should be given, we added correlational results related to a level of family support. They were presented in Table 1, showing that family support negatively correlated with psychological, social, and total pain, and stress, but positively correlated with self-efficacy.
